# The Effect of Smog-Related Factors on Korean Domestic Tourists’ Decision-Making Process

**DOI:** 10.3390/ijerph17103706

**Published:** 2020-05-25

**Authors:** JunHui Wang, JooHyang Kim, JiHyo Moon, HakJun Song

**Affiliations:** 1College of Tourism and Fashion, Pai Chai University, Daejeon 35345, Korea; 528734937qq@gmail.com; 2Department of Convention and Hotel Management, Hannam University, Daejeon 34430, Korea; jhk3456@hnu.kr; 3School of Business, Hanyang University, Seoul 133-791, Korea; tourism88@naver.com

**Keywords:** threat, protection motivation for smog, policy, extended model of goal-directed behavior, structural equation modeling, tourists’ behavior

## Abstract

The present study aims to explore Korean domestic tourists’ decision-making processes by utilizing an extended model of goal-directed behavior (EMGB) as a theoretical framework. Integrating government policy (PLY) and protection motivation for smog (PMS) with the original model of goal-directed behavior (MGB) makes it easier to better understand the formation process of tourists’ behavioral intentions for domestic travel. Structural equation modeling (SEM) is employed to identify the structural relationships among the latent variables. The results of the EMGB indicated that desire had the strongest effect on the behavioral intention of tourists to travel domestically; positive anticipated emotion is the main source of desire, followed by negative anticipated emotion. Government PLY on smog has a significant, positive and indirect effect on behavioral intentions of domestic or potential tourists through the protection motive theory. We found that desires are verified as a determinant of the behavioral intention’s formation, more significant than that of perceived behavioral control, frequency of past behavior and protection motivation. In addition, this study offers theoretical and practical suggestions.

## 1. Introduction

The dangers of global environmental changes are expanding at a rapid pace. This environmental threat has transformed into a serious issue, as well as disease and political issues have become an important factor hindering the development of tourism [1,2,3]. Among various environmental issues, there are a few studies on the effect of air pollution, including smog pollution (i.e., particulate matter (PM) 2.5 and 10) [4,5,6]. Consistent with this change is the threat of smog. It is essential that tourists are assured of appropriate safeguards through the government’s policy (PLY) and their protection motivation for smog (PMS), to protect them from the harm of smog. Recognizing the need for the tourism industry to have measures to minimize the influence of smog on tourists, tourism managers and operators need to consider tourists’ emerging requests about smog into their marketing strategies. Specifically, they seem to be interested in handling their tourism businesses in a healthy manner and in implementing social health practices against the threat of smog.

However, research using a theoretical framework to understand tourists’ behavioral intentions in the situation of smog has possibly been ignored. One appropriate way of addressing this shortcoming and developing the knowledge of tourists’ behaviors in relation to smog is to search for a proper theoretical framework. Employing an effective theoretical framework for tourists can provide a worthwhile understanding of the complicated process by which tourists generate their behavioral intentions regarding smog. Although it is not an easy task to understand the complex decision-making processes of tourists with regard to the threat of smog, employing the concept of behavioral intention to travel can be a useful resource to determine their decision-making processes. Tourists usually have shaped behavioral intentions from their own thought processes and behavioral intention has played a vital role in forming actual tourism behavior. In this regard, the recent social psychological theory to understand human behavioral intention, model of goal-directed behavior (MGB) [7], can be applied to this situation to elucidate the behavioral intention of tourists toward smog. Due to its higher predictive ability, MGB has been employed to understand various tourist behaviors. Despite the effectiveness of the MGB, it has not been introduced in the context of domestic tourist behaviors toward smog. Therefore, the current study aims to explore tourists’ behavioral intentions using an extended model of goal-directed behavior (EMGB) that adds government PLY and individual protection motivation to the original MGB. We hypothesized that under the threat of smog, potential domestic tourists would contemplate PMS and are likely to be influenced by the government PLY that deals with smog, which could lessen their risk of smog exposure while traveling.

According to the PLY, tourists responded by increasing their behavioral intention for domestic travel despite the danger of smog because the PLY can create an atmosphere for tourists to be comparatively safe from smog while also enhancing the level of PMS by increasing its effectiveness. The PLY usually composed of building space facilities (e.g., indoor sports facilities and air purification facilities), tightening the emission standards of coal-fired power plants and imposing emission charges, and pinpointing the cause of smog. Moreover, tourists may take the PMS due to limited information and facilities to protect themselves from the threat of smog. The PMS can include discussions with other people about protection from smog before travel, collecting news and information on policies about smog, and planning a travel time to avoid smog. Using a sample of Korean potential domestic tourists, we try to identify tourists’ decision-making procedures by applying the MGB theory as well as a model that additionally contains the PLY and the PMS that responds to the smog. The present research has several aims. First, we examine potential travelers’ decision-making processes when the risk of smog discourages travel, with an EMGB explaining travel intent that includes the PLY and the PMS. Second, we investigate the PLY and the PMS toward smog in a sample of Korean potential travelers and the performance of these constructs in a proposed theoretical framework. Finally, we present effective implications for the government PLY and the PMS, which contribute to mitigating the negative perception of smog and benefit the public and tourism businesses such as tourism marketers, hospitality services, transport systems and government agencies.

## 2. Literature Review

### 2.1. MGB and EMGB

To better understand the formation process of personal behavior intentions, based on the theory of planned behavior (TPB) and the theory of reasoned action (TRA), Perugini and Bagozzi [7] added anticipated emotional factors, past behaviors and desires to construct the MGB theoretical framework. In detail, desire is a kind of motivation that determines individual behavior. The anticipated emotional factors (positive and negative) expected for a particular behavior may be important factors that impact desire and behavioral intentions in the decision-making processes. Past behavior may be an imperative factor that affects desire and behavior intention. The addition of these factors not only solves the limitations of the TPB and TRA, but also significantly enhances the explanatory power of individual decision-making processes, thus making complex individual decision-making behavior processes easier to interpret.

As a theoretical basis, the MGB has been widely used in tourism research for various tourism and leisure behaviors. Kim et al. [8] confirmed the moderating role of gender differences in overseas travel decision-making processes based on the MGB theory. The results verified that various factors (i.e., attitude, subjective norm, perceived behavioral control and anticipated emotions) have a significant effect on behavioral intention through mediating variable desire. Past behavior also has a positive effect on behavioral intention. In the context of South Korea’s popular culture in Asian countries, Lee et al. [9] conducted research on Chinese pop cultural fans by combining the MGB model and the Attention/Awareness, Interest, desire and Action (AIDA) model. The results showed that the intention to travel to South Korea (hereafter Korea) for pop culture was influenced by anticipated emotions, subjective norms, perceived behavioral control and desire. Notably, desire has the most significant effect on behavioral intentions, consistent with the findings of Perugini and Bagozzi [10]. Choe, Kim and Cho [11] utilized the MGB to explain the effect of patriotism on behavioral intentions (e.g., event attendance intention and media consumption intention) in the context of Rio 2016, which verified not only the indirect influence of factors such as attitude, subjective norms, positive and negative anticipated emotions, perceived behavioral control and desire on behavioral intentions, but also the significant direct impact of patriotism on media consumption intentions.

In addition, Ajzen [12] and Perugini and Bagozzi [13] believed that to further improve the theory’s ability to predict the behavioral intention of a specific group of people, and better understand the psychological state of the behavior decision process, it should consider new constructs or changing relationships among factors to further develop the EMGB under different conditions. Based on this research, in the field of tourism research (i.e., festivals, casinos, hotels, slow tourism, museums, etc.), many scholars have adopted the EMGB theory to explain tourist decision-making processes under different backgrounds and different conditions [8,14,15,16,17,18,19]. Song et al. [17] combined environmentally friendly tourism behaviors with the MGB model to explore factors that influence the revisit intention of tourists at Korea’s Boryeong Mud Festival. The results showed that in the EMGB, attitudes, subjective norms, positive anticipated emotions, frequency of past behavior and environmentally friendly tourism behaviors affect behavioral intentions by influencing desires. Song et al. [20] added two new constructs of image and perception related to the Oriental Medicine Festival into the MGB and investigated the formation process of the revisit intention of the festival. The study validated the effects of attitudes, subjective norms, positive anticipated emotions on desire, and subsequently, on revisit behavioral intentions. The factors of image and perception have a significant positive effect on the attitude of participating in the Oriental Medicine Festival. Han and Yoon [15] reconstructed the EMGB to better explain the customer’s decision-making processes when choosing an environmentally responsible hotel by adding environmental awareness, perceived effectiveness, eco-friendly behavior and reputation into the MGB. The relationships among the variables have also been effectively verified. Han, Kim and Lee [14] integrated problem awareness, affective commitment and non-green alternative attractiveness into the MGB and examined the decision-making process for environmentally responsible museums. They also verified the significant effect among variables in the original MGB, which is almost the same as with previous studies. Meng and Choi [16] stated that integrating perception of authenticity, the EMGB can better elucidate tourists’ intention to visit a slow tourism destination. As a result, attitude, subjective norm, positive and negative anticipated emotion and perceived behavioral control were verified as factors influencing desires and intentions. The frequency of past behavior significantly and directly impacts desires and intentions. The perception of authenticity directly influences desires. By integrating mass media impact and perception of climate change, Kim et al. [8] established an EMGB model to study the impact of climate change on the decision-making processes of potential tourists overseas. The results showed that positive anticipated emotion has the most important impact on the desire for overseas travel, followed by attitude, subjective norm and perceived behavioral control. desire affects behavioral intention more significantly than perceived behavior control, perception of climate change and mass media impact. Thus, based on the aforementioned studies, the following hypotheses were suggested:
**H1.** *Attitude has a positive influence on**the desire of domestic tourists;*
**H2.** *Subjective norm has a positive influence on**the desire of domestic tourists;*
**H3.** *Positive anticipated emotion has a positive influence on**the desire of domestic tourists;*
**H4.** Negative anticipated emotion has a positive influence on the desire of domestic tourists;
**H5.** Perceived behavioral control has a positive influence on the desire of domestic tourists;
**H6.** Frequency of past behavior has a positive influence on the desire of domestic tourists;
**H7.** *Perceived behavioral control has a positive influence on**the behavioral intention of domestic tourists;*
**H8.** Frequency of past behavior has a positive influence on the behavioral intention of domestic tourists;
**H9.** Desire has a positive influence on the behavioral intention of domestic tourists.

### 2.2. Korean Government Policy for Smog

Few studies have been published on the impact of government policies on individual behavior. Lee, Jung and Kim [21] examined the effect of a policy intervention that provides an upper limit for handset subsidies on users’ intention to change handset and household expenses on mobile telecommunications. Sun et al. [22] conducted a questionnaire survey in Beijing, China and found that the policy of “No traffic restrictions for EVs (electric vehicles)” significantly affected the users’ opinions on using EVs and adoption intention. Wang, Li and Zhao [23] divided policy measures into three categories (i.e., financial incentive policy measures, information provision policy measures and convenience policy measures) and investigated how these policy measures motivate consumers to adopt EVs and how such effects are moderated by consumers’ environmental concerns.

Recently, the necessity of research on smog has increased because it is an important issue for tourism practitioners and policy makers. In this regard, the effect of government policy on smog, especially in tourism, is imperative. In the context of tourism, the government policy has been mainly emphasized by the increase in social welfare by sustaining a safe environment for both demanders and suppliers. Due to the increased concern about smog, the effect of policy is no longer limited to a single industry. As the effect of tourism policy affects travel agencies, hotels and tourism transportation, various policies need to be implemented in the situation of smog. Smog policy should be considered since it can either larger or minimize the negative effect of smog on the tourism industry. Moreover, policies to reduce the negative effect of smog can change not only individual tourism behavior, but also various tourism-related businesses. In the present study, we plan to scrutinize the influence of government policies on the intention to undertake domestic travel during the threat of smog.

For the empirical analysis, Korea’s management policy for smog was chosen. The policy was planned to reduce the danger of smog on people’s outdoor activities, including travel and tourism, by building several indoor sports or air purification facilities, imposing emission charges for areas or companies that generate heavy smog and tracking the cause of smog. The current study analyzes the effect of the PLY on smog to offer insights into potential tourists, tourism practitioners and government agencies. Although the Korean tourism industry is known for its progressive infrastructure, devices and services, it has inherent problems. Despite the significant impact of smog on Korean society, there is a lack of studies that identify government policy for tackling smog and its effect on tourists’ decision-making processes.

Thus, the following hypotheses were suggested:
**H10.** Policy has a positive influence on the desire of domestic tourists.
**H11.** Policy has a positive influence on the behavioral intention of domestic tourists.

### 2.3. Protection Motivation for Smog

In terms of PMS, a related theory called the protection motivation theory (PMT) should be considered. The PMT explains what individuals decide and why they take protection action [24,25]. In other words, the PMT describes how decision-makers can generate adequate protection behaviors against threats. This may include reducing or avoiding activities, thereby changing the entire decision-making process and prompting decision-makers to make new decisions. The core of the PMT is threat appraisal and coping appraisal, in which threat appraisal is people’s understanding of threat factors or perceived severity and vulnerability. Coping appraisal is an individual’s ability to cope with and avoid dangers or response effectiveness and self-efficacy. When individuals perceive the severity and likelihood of risk and the availability of effective means of controlling consequences, including external actions and the individual’s own ability, the likelihood of engaging in protective behavior is high [26]. The PMT is mainly used to explain people’s decision to participate in health risk reduction behaviors and environmental behaviors [27,28,29,30,31]. Janmaimool [27] stated that individuals will take individual precautionary decisions to minimize risks based on perceived risk vulnerability and the severity of the negative consequences. When people have higher perceived severity, vulnerability and higher perceived self-efficacy and response effectiveness, they may participate in environmental protection behaviors to minimize risks. Kim et al. [29] and Marquit [30] found that individuals’ intentions to participate in environmental protection behaviors were significantly influenced by the attributes of protection motivation. Tsai et al. [32] provided a theoretical framework for the security protection of Internet users and found that when Internet users encounter a variety of online security threats, they evaluate the threats and download security software to take precautions to protect personal online safety to avoid loss. Le and Arcodia [33] found that when tourists notice vacation-related risks based on their subjective risk perception and past travel experience, it is easy to take protective behaviors [26].

However, in numerous studies in the tourism industry, the application of PMT is not widespread, with the exception of the following related studies that were verified. Verkoeyen & Nepal [34] help to understand the impact of coral bleaching on the behavioral intentions of diving tourists by applying the PMT. Among them, PMT can explain 12.8% to 47.7% of the variance in behavioral intentions. Wang et al. [35] highlighted the impact of climate change on the attractiveness of coastal tourism destinations and tourist behavior intentions, of which PMT explained 58% of the variation in tourist behavior intentions. Horng et al. [36] also found that PMT can explain 43.7% of tourism energy saving and carbon reduction behavior intentions variation of Asian tourists. Therefore, this study introduces the PMS into the research framework of domestic tourists’ decision-making processes under the influence of smog and further analyzes the behavior of domestic tourists. As far as this article is concerned, PMS refers to people’s perception of how serious the smog is, how vulnerable individuals are and the assessment of individuals’ abilities to cope with it. Based on the aforementioned research, the following hypotheses are proposed:
**H12.** PMS has a positive impact on the desire of domestic tourists;
**H13.** PMS has a positive impact on the behavioral intentions of domestic tourists;
**H14.** Policy has a positive impact on PMS.

## 3. Methodology

Based on previous studies, the measurement items used in the present study were created. For the original MGB model, it contains a total of 32 items recommended by previous studies [14,17,18,19], including attitude composed of 4 items (e.g., “I think domestic travel is beneficial”), subjective norm with 4 items (e.g., “Those who influence my decision will support me for domestic travel”), perceived behavior control was operationalized with 3 items (e.g., “I can travel domestically at any time as I want”), positive anticipated emotion with 4 items (e.g., “I will be excited if I can travel domestically”), negative anticipated emotion with 4 items (e.g., “I will be angry if I cannot travel domestically”), desire with 4 items (e.g., “I want to travel domestically in the near future”), behavioral intention with 4 items (e.g., “I will make an effort to travel domestically in the near future”), frequency of past behavior with 1 item, policy with 4 items (e.g., “The government is pinpointing the cause of smog”) and protect motivation of smog with 3 items (e.g., “I searched for news and policies related to smog in Korea before traveling domestically”). All the survey items were measured using a 5-point Likert scale (1 = “strongly disagree” to 5 = “strongly agree”).

The study’s selected subjects were adults who lived in Korea and had experienced domestic travel within the past year. To collect data, the Embrain Research panel, which has the largest number of panelists in Korea, was used for the efficiency of recruiting research targets and collecting data suitable for research purposes. Since the company has approximately 1.3 million national panelists suitable for the census of the National Statistical Office, it is believed that the representativeness of the sample is sufficiently secured. In this online survey, the target number of respondents was determined as almost 600, considering the minimum sample count, panelist response rate and median dropout rate. Specifically, 611 questionnaires were collected through the allocation sample extraction by referring to the quota table for gender and population proportions by region provided by the Korea National Statistical Office. As a result of excluding invalid responses and dropouts, 583 questionnaires were finally used for the empirical analysis. The survey was conducted online from 1–15 May 2019. A link to the survey was sent via an email that the panel registered to the research company, along with a description of the purpose of the investigation, the time it took to respond and the compensation. To participate in this survey, respondents voluntarily clicked the “Join Survey” button to proceed. Descriptive statistics and SEM were carried out using R-studio.

## 4. Results

### 4.1. Descriptive Statistics

The characteristics of the respondents are reported in Table 1. From the 583 respondents, 48.7% were male and 52.3% were female; 23.7% were between the ages of 40–49, 22.8% were between the ages of 50–59 and 67.2% were married. In terms of academic qualifications, 58.8% of respondents were enrolled or graduated from a four-year university program, 26.4% were technicians or academics, 16.5% were housewives and 12.2% were civil workers. In terms of monthly average income, 21.9% of respondents’ income level ranged between KRW 2–2.9 million and 19.4% between KRW 3–3.9 million.

### 4.2. Measurement Model

The results of the measurement model revealed a satisfactory model fit to the data: χ^2^ (491) = 868.522, *p* < 0.001, CFI (Comparative Fit Index) = 0.967, TLI (Tucker-Lewis Index) = 0.927, NNFI (Non-Normed Fit index) = 0.962 and RMSEA (Root Mean Square Error of Approximation) = 0.036. The value of TLI and CFI over 0.9 and RMSEA below 0.05 confirms a good model fit [37]. The standardized factor loadings of the observed variables ranged from 0.655 to 0.962 (see Table 2), which exceeded or were close to the ideal criterion of 0.7. The average variance extracted (AVE) values were above the recommended value of 0.5 [10] (see Table 3) and composite reliability values for the multi-item scales were larger than the minimum criteria of 0.7. As reported in Table 3, AVE can be used to check the discriminant validity of constructs in the measurement model [38] and all AVEs of each construct should be larger than the squared correlation to demonstrate satisfactory discriminant validity. Thus, it was confirmed that the proposed measurement model fit the data well.

### 4.3. Structural Model

After identifying a well-fitted measurement model, the relationships between all latent variables in the structural model were tested. The results of the structural model showed an excellent fit to the data (χ^2^ = 976.230, *df* = 532, *p* < 0.001, χ^2^/*df* = 1.835, RMSEA = 0.038, CFI = 0.962, TLI = 0.920, NNFI = 0.957) (see Table 4 and Figure 1). The results of the EMGB show that positive anticipated emotions (β_PAE__→DE_ = 0.612, *t* = 11.837, *p* < 0.001) and negative anticipated emotion (β_NAE→DE_ = 0.194, *t* = 5.272, *p* < 0.001) have a significant influence on desire to travel domestically, supporting H3 and H4. However, attitudes (β_ATT→DE_ = 0.075, *t* = 1.242, not significant), subjective norms (β_SN→DE_ = 0.043, *t* = 0.833, not significant), perceived behavioral control (β_PBC→DE_ = 0.065, *t* = 1.761, not significant), the frequency of past behavior (β_FPB→DE_ = −0.017, *t* = −0.639, not significant), PLY (β_PLY→DE_ = −0.013, *t* = −0.359, not significant) and PMS (β_PMS→DE_ = 0.006, *t* = 0.181, not significant) are not statistically significant in predicting the desire to travel domestically. It also shows that perceived behavioral control (β_PBC→BI_ = 0.079, *t* = 2.040, *p* < 0.05), desire (β_DE→BI_ = 0.745, *t* = 19.600, *p* < 0.001), frequency of past behavior and perception of smog (β_FPB→BI_ = 0.071, *t* = 2.659, *p* < 0.01) and PMS (β_PMS→BI_ = 0.082, *t* = 2.648, *p* < 0.01) have significant effects on behavioral intention to travel domestically, supporting H7, H8, H9 and H13. In addition, policies could also influence behavioral intention indirectly through PMS as a mediator (β_PLY→PM_ = 0.323, *t* = 6.783, *p* < 0.001), supporting H14.

## 5. Discussion and Limitation

### 5.1. Discussion

Kiatkawsin and Han [39] and Trang, Lee and Han [40] pointed out that environmental quality impacts the attractiveness, sustainability and competitiveness of tourism destinations because it is considered a core factor that affects tourists’ decision-making processes. As an important aspect of environmental quality, air pollution, especially smog, has become an important factor influencing potential tourists’ domestic travel intentions [1,35]. Explaining the psychology of Korean tourists’ domestic travel intentions under the threat of smog and the formation process of travel intentions more thoroughly, this study has a guiding significance for tourism marketers and developers [41]. The government policies and PMS in this article are added to the MGB framework to further explore the domestic travel decision-making process of tourists under the threat of smog. Finally, a conceptual research model containing 14 hypotheses was proposed, which was verified based on the results of a questionnaire survey administered to 583 respondents. The analysis results showed that seven hypotheses were accepted and seven were rejected.

Consistent with previous studies [7,12], adding new constructs or changing relationships among factors should be considered to further develop the EMGB under different conditions to improve the MGB’s ability to predict the behavioral intention of a specific group of people. As mentioned in the literature review, scholars have integrated necessary variables into the original MGB to better explain the behavior of specific groups under different conditions [8,14,15,16,17,18,19]. The results of the present study verified that two additional constructs (policy and PMS) also significantly influence domestic tourists’ behavioral intention under the threat of smog in Korea. The EMGB also helps to avoid possible misspecification, including ignoring important variables or considering unimportant variables. In other words, the two new structures of policy and protection motivation make it easier to understand the complex psychology of Korean tourists in making domestic tourism decisions under the threat of smog.

Contrary to results of previous studies, anticipated emotions (positive and negative emotions) are verified as the only source of desire, and thus affect behavioral intentions [16,17,20]. There was no significant influence between attitude, subjective norms, perceived behavioral control and desire. Please note that this is only a relative concept, and it does not mean that attitude, subjective intention perceived behavior control have no effect on desire, but that in the original MGB framework. It was also found that anticipated emotions had the strongest influence on desire, explaining the source of 66.4% desire. Therefore, positive or negative anticipation emotions make tourists eager to travel domestically. In other words, the desire to travel is dependent only on whether the individual is happy or not. In this process, it was confirmed that the value judgment (attitude) of domestic travel, opinions of the people around (subjective norms), and the personal financial and material resources (perceived behavior control) can be ignored here. Consistent with Kim et al.’s [8] study, which also verified that positive anticipated emotion has the most important impact on the desire toward overseas travel compared with other variables in the context of climate change. The current study is the first trial in existing research on whether government policies and PMS have been successfully incorporated into the original MGB framework. In the empirical results, it is worth noting that government policies will only have a positive and meaningful impact on behavioral intentions through the intermediate role of PMS. Thus, policies could have a significant impact on tourist protection psychology. The implementation of policies can psychologically reduce the internal insecurity, anxiety and worry caused by smog in domestic tourists. The implementation of powerful smog-related policy measures can also create a less polluted and healthier domestic tourism environment for domestic tourists from a practical perspective. In addition, the disclosure of government policies and information also helps tourists who are eager to travel domestically obtain information faster and more effectively, avoid risks and protect themselves as much as possible. The PMS based on personal protection psychology directly affects the behavior intentions of individuals in this study. This result agrees with previous studies [33,34,35,36]; for example, Le and Arcodia [33] stated that when tourists notice vacation-related risks based on their subjective risk perception and past travel experience, it is easy to take protective behaviors. Although the PMT could explain how decision-makers can generate adequate protection behaviors, including avoiding activities, changing the entire making process or making new decisions in the face of threats, the application of the PMT is not widespread in tourism industry studies. Moreover, it was identified that desire could affect behavioral intention more significantly than perceived behavior control, PMS and frequency of past behavior, which was also consistent with previous studies [8,14,15,16,17,18,19].

As revealed by the present study, anticipated emotions (positive and negative emotions) are the only sources of desire, as demonstrated by 66.4% of the sample. Therefore, tourism industry practitioners and developers must pay attention to the current background of smog and determine how to stimulate the hearts of tourists, enhance attractiveness and encourage domestic travel.

### 5.2. Limitation

This study has some limitations, which should be considered in future research.

First, smog occurs seasonally; its severity differs as per the seasons and the results of the survey will thus be different. As the present study’s survey was conducted between 1–15 May 2019 and the degree of smog varied daily, these results can only represent the perception state of tourists during that period and not the perception of tourists in all periods.

Second, this study only focuses on Korean domestic tourists’ intentions under the threat of smog. It is also possible to further consider international tourists as the object of investigation. It may be interesting to compare results of studies on international tourists and domestic tourists to determine which tourists are more sensitive to smog, as this could help in making travel plans more suitable for tourists.

Finally, future research could also make full use of the EMGB by adding more specific factors to the original model. For example, mass media will also be an important factor influencing tourists’ decision-making processes under the threat of smog, considering the impact of mass media could help better understand the formation process of tourists’ decision-making behavior.

## 6. Conclusions

As modern society develops, the issue of environmental pollution is becoming serious and its influence is expanding to the tourism field. Therefore, the issue of the effect of smog on the decision-making processes of tourists is no longer a problem for one country alone. As the impact of smog is expected to intensify in the future, this study can serve as an important basic study to demonstrate how tourism practitioners in other countries can cope with a possible smog situation in the future. A future study examining how these results can be applied in other countries will be meaningful from an academic and practical perspective.

Based on these results, certain suggestions are offered:

Media’s storytelling: Attract domestic tourists with tourism promotion programs that are interesting and tell stories. With the advent of the self-media era, content and methods of information dissemination have significantly changed. Instead of relying on a single newspaper, media, offline publicity and other channels, the rapid development of self-media short video platforms such as YouTube or TikTok has induced tremendous changes in content and methods of information dissemination, through the design of storyline, recording, editing, dubbing, sharing, and other processes to make tourism publicity more attractive. The platform’s comment and recommendation functions also enable more people to pay close attention to issues such as tourist destination image, attractiveness, information and service quality through the media platform.

Private customization: Design and develop domestic tourism products or routes that meet the inner needs of tourists. Travel agencies should plan and design to appeal to tourists by providing them unique and attractive travel experiences. Higher anticipated emotions can stimulate tourists to travel domestically as soon as possible. As far as domestic tourism in the background of smog is concerned, it is necessary to choose tourist destinations which have good environmental quality and are beneficial to the physical and mental health of tourists.

Government support: To stimulate domestic travel, the Korean government must accelerate the implementation of smog control measures. It must also promptly publish smog-related information to allow tourists to know the environmental status of tourist destinations to enable them to make reasonable choices and try to avoid destinations that are impacted by smog. Li et al. [4] stated that the spread of information about smog by the mass media will not only worsen or undermine the established tourists’ impression of the city, but also lead to the shrinking of the potential tourism market and the loss of tourism revenue. Thus, the government has a restrictive role on the mass media and attempts to ensure the authenticity and validity of smog-related information to avoid misinformation that can cause psychological panic among tourists or potential tourists. Moreover, investigating the causes of the smog; improving or building facilities that can protect against smog; tightening emission standards of coal-fired power plants; and levying emission fees to reduce smog emissions and encourage the promotion of green cars can provide tourists with a safe and healthy domestic travel environment and thus promote domestic travel decisions.

## Figures and Tables

**Figure 1 ijerph-17-03706-f001:**
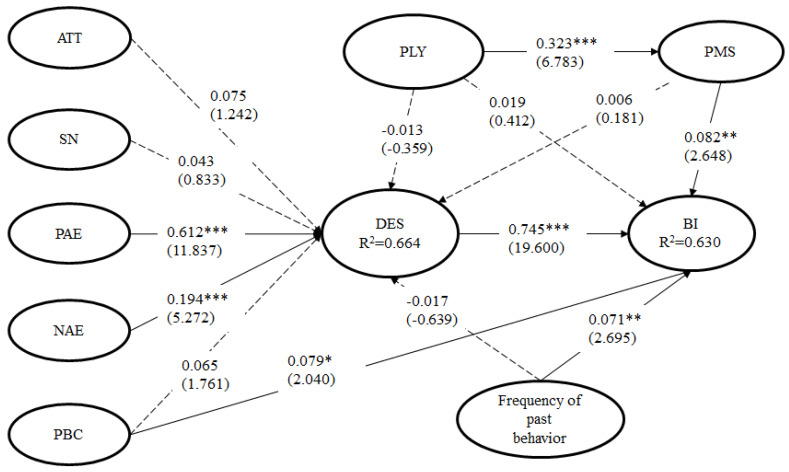
Structural model results. Notes 1: * *p* < 0.05; ** *p* < 0.01; *** *p* < 0.001. Notes 2: ATT = attitude; SN = subjective norm; PAE = positive anticipated emotion; NAE = negative anticipated emotion; PBC = perceived behavioral control; PLY = policy; PMS = protection motivation for smog; DES = desire; BI = behavioral intention.

**Table 1 ijerph-17-03706-t001:** Profile of survey respondents (*n* = 583).

	Characteristic	Frequency	Percentage (%)
Gender	Male	284	48.7
	Female	299	52.3
Age	20–29	102	17.5
	30–39	105	18.0
	40–49	138	23.7
	50–59	133	22.8
	>60	105	18.0
Education	High school or less	100	17.2
	College	69	11.8
	Four-year University	343	58.8
	Graduate school	71	12.2
Occupation	Technician/Academic	154	26.4
	Business person or self-employed	42	7.2
	Service person	69	11.8
	Office worker	71	12.2
	Civil workers	27	4.6
	Student	47	8.1
	Housewives	96	16.5
	Freelancer	26	4.5
	Retired	17	2.9
	Others	34	5.8
Marital status	Single	181	31.1
	Married	392	67.2
	Others	10	1.7
Monthly income	Less than KRW1 million	54	9.3
	KRW 1–1.9 million	76	13.0
	KRW 2–2.9 million	128	21.9
	KRW 3–3.9 million	113	19.4
	KRW 4–4.9 million	77	13.2
	KRW 5–5.9 million	54	9.3
	KRW 6–6.9 million	28	4.8
	KRW 7–7.9 million	22	3.8
	Over KRW 8 million	31	5.3

**Table 2 ijerph-17-03706-t002:** Confirmatory factor analysis and factor loadings.

Nine Factors and Scale Items	Standardized Loading
**F1: Attitude**	
I think domestic travel is positive.	0.863
I think domestic travel is beneficial.	0.907
I think domestic travel is very valuable.	0.896
I think domestic travel is attractive.	0.853
**F2: Subjective norm**	
Those who influence my decision will approve me for domestic travel.	0.859
Those who influence my decision will support me for domestic travel.	0.902
Those who influence my decision will understand my domestic travel.	0.875
Those who influence my decision will recommend me for a domestic travel.	0.870
**F3: Perceived behavioral control**	
I can travel domestically at any time I want.	0.721
I have the overall ability to travel domestically.	0.917
I have enough financial resources to travel domestically.	0.828
**F4: Positive anticipated emotion**	
I will be excited if I can travel domestically.	0.905
I will be glad if I can travel domestically.	0.889
I will be satisfied if I can travel domestically.	0.870
I will be happy if I can travel domestically.	0.925
**F5: Negative anticipated emotion**	
I will be angry if I cannot travel domestically.	0.866
I will be disappointed if I cannot travel domestically.	0.920
I will be worried if I cannot travel domestically.	0.759
I will be upset if I cannot travel domestically.	0.859
**F6: policy**	
The government is improving or building active space facilities (such as indoor sports facilities, air purification facilities, etc.) that can protect against smog.	0.716
To reduce smog emissions, the government has tightened emission standards of coal-fired power plants and imposed emission charges.	0.733
The government is pinpointing the cause of smog.	0.777
The government is pursuing a policy to spread green cars.	0.784
**F7: Protection motivation for smog**	
I considered discussing with the people around me to protect myself from smog before traveling domestically.	0.788
I searched for news and policies related to smog in South Korea before traveling domestically.	0.825
I planned the best tourist route to avoid smog before traveling domestically.	0.844
**F8: Desire**	
I am interested in traveling domestically in the near future.	0.864
I want to travel domestically in the near future.	0.908
I hope to travel domestically in the near future.	0.911
I am eager to travel domestically in the near future.	0.823
**F9: Behavioral intention**	
I am planning to travel domestically in the near future.	0.830
I will make an effort to travel domestically in the near future.	0.858
I would like to travel domestically again in the near future.	0.867
I am willing to invest money and time for domestic travel in the near future.	0.908

Notes: All standardized factor loadings are significant at *p* < 0.001.

**Table 3 ijerph-17-03706-t003:** Results of measurement model.

Construct	ATT	SN	PAE	NAE	PBC	PLY	PMS	DES	BI
ATT	**0.774**	0.548	0.530	0.091	0.125	0.018	0.064	0.395	0.300
		(0.741)	(0.728)	(0.301)	(0.353)	(0.134)	(0.253)	(0.629)	(0.547)
SN	0.028	**0.769**	0.401	0.079	0.163	0.025	0.065	0.315	0.257
			(0.634)	(0.281)	(0.404)	(0.157)	(0.256)	(0.561)	(.507)
PAE	0.026	0.026	**0.806**	0.117	0.142	0.010	0.023	0.606	0.442
				(0.342)	(0.377)	(0.100)	(0.150)	(0.778)	(0.665)
NAE	0.025	0.023	0.026	**0.728**	0.062	0.076	0.087	0.201	0.166
					(0.249)	(0.276)	(0.295)	(0.448)	(0.407)
PBC	0.023	0.021	0.021	0.026	**0.683**	0.043	0.070	0.148	0.156
						(0.209)	(0.264)	(0.385)	(0.395)
PLY	0.027	0.024	0.024	0.034	0.027	**0.567**	0.091	0.018	0.025
							(0.302)	(0.134)	(0.158)
PMS	0.023	0.022	0.022	0.029	0.024	0.029	**0.671**	0.039	0.067
								(0.197)	**(0.258)**
DES	0.025	0.023	0.028	0.026	0.021	0.024	0.020	**0.769**	**0.610 ***
									**(0.781)**
BI	0.024	0.024	0.027	0.028	0.021	0.025	0.021	0.029	0.750
**α**	0.930	0.929	0.942	0.913	0.856	0.838	0.859	0.926	0.923
**CR**	0.932	0.930	0.943	0.914	0.865	0.839	0.860	0.930	0.923

*Note*: ATT = attitude; SN = subjective norm; PAE = positive anticipated emotion; NAE = negative anticipated emotion; PBC = perceived behavioral control; PLY = policy; PMS = protection motivation for smog; DES = desire; BI = behavioral intention; CR = composite reliability; * Highest correlation between pairs of constructs; AVE values are along the diagonal; squared correlations among latent constructs are above the diagonal; correlations among latent constructs are in parentheses; standard errors among latent constructs are below the diagonal.

**Table 4 ijerph-17-03706-t004:** Standardized parameter estimates of structural model.

	Hypotheses	Coefficients	*t-*Value	Test of Hypotheses
H1	ATT_→_DE	0.075	1.242	Rejected
H2	SN_→_DE	0.043	0.833	Rejected
H3	PAE_→_DE	0.612	11.837	Accepted
H4	NAE_→_DE	0.194	5.272	Accepted
H5	PBC_→_DE	0.065	1.761	Rejected
H6	FPB_→_DE	−0.017	−0.639	Rejected
H7	PBC_→_BI	0.079	2.040	Accepted
H8	FPB_→_BI	0.071	2.695	Accepted
H9	DE_→_BI	0.745	19.600	Accepted
H10	PLY_→_DE	−0.013	−0.359	Rejected
H11	PLY_→_BI	0.323	6.783	Accepted
H12	PMS_→_DE	0.006	0.181	Rejected
H13	PMS_→_BI	0.082	2.648	Accepted
H14	PLY_→_PMS	0.019	0.412	Rejected

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
