# Peer review of "The Effect of Smog-Related Factors on Korean Domestic Tourists’ Decision-Making Process"

_ijerph, 2020, doi:10.3390/ijerph17103706_

Round 1

Reviewer 1 Report

The subject of the article is interesting and it is linked to the objectives of the journal, however, there are a number of issues that have to be reconsidered.

For a better visibility on databases, the authors are asked not to repeat among keyword the words/concepts included on the title of the article.

lines 34-35. Abbreviation as M2.5 and PM10 should be exlained, e.g. Particulate Matter. Please keep that in mind for the entire article

3.2. Data Collection. Ait is necessary a deeepr explanation for explainig why the 683 companies are resentative for the intire studied population.

It is recommended to split Discussions and Conclusion in two disctincte parts.

Also, it is necessray to better mention the limits of the research.

Reviewer 2 Report

This is a straightforward paper addressing a concrete domain.  The modelling (adaptation) process is clearly outlined and the variables explained.  The results are interesting (although not really surprising for me - having researched in the domain of tourism for the last 20 years).  The hypotheses testing (SEM results) are clearly presented and discussed.  On that basis, the authors draw conlcusions and recommendations directly related to their findings.  All in all, a solid piece of work.  There are only two issues, I want to address.  First, the 'outsourcing' of data collection to an external entity, without at least outlining any methodological / structural guarantees for reliability / bias in the data collection process.  The second issue is language.  The paper, needs some heavy language and grammar editing.

Reviewer 3 Report

This paper deals with a highly-important topic, and its results are internationally important. The paper is based on the both theoretical and empirical bases. It is generally well-written,. I recommend it for acceptance after certain revisions. See my recommendations below.

  • The number of sources cited in the whole work and, particularly, Introduction should be increased.
  • Methods -> Methodology. I also suggest to extend this section and to avoid splitting it into two sub-sections. IMPORTANT: I cannot understand whether the authors themselves collected the data and, if not, whether the authors have permission to use these data. Clarifications MUST exist to avoid ethical issues.
  • Data analysis -> Results
  • Table 2 should have a more clear title.
  • Conclusion and Discussion -> Discussion and Conclusion. Please, start this section with interpretations and end it with general conclusions and perspectives for further research.
  • The readers need a table showing which hypotheses are verified and which not. Also, please, relate factors from Table 2 to the hypotheses anyhow (e.g., on additional figure).
  • Line 353: Different from the previous research... Please, give citations.
  • Discussion should also explain briefly what this example tells to tourism managers and planners from the other countries.
  • The writing is clear, but I encourage the authors to polish the language a bit else. I also suggest to write the abstract in Present tense, not in Past tense.
  • I strongly encourage the authors to choose the other type for this paper – Communication instead of Article. In this case, the brevity of this paper becomes a kind of advantage, and it is suitable.

Good luck with revision!

Round 2

Reviewer 1 Report

The arctile was improved. it answered to my suggestions.